Report

# Self-organization of plasticity and specialization in a primitively social insect

## Authors

Solenn Patalano, Adolfo Alsina,
Carlos Gregorio-Rodríguez, ...,
Seirian Sumner, Wolf Reik,
Steffen Rulands

## Correspondence

patalano@fleming.gr (S.P.),
wreik@altoslabs.com (W.R.),
rulands@pks.mpg.de (S.R.)

## Graphical abstract

## In brief

Combining theory with experiments, Patalano et al. show that both robust specialization and rapid plasticity in primitive social wasps result from a self-organized balance between the development of a molecular queen phenotype and its colony-scale inhibition. This allows *Polistes* to be stable against intrinsic molecular perturbations while reacting plastically to extrinsic cues.

## Highlights

- Experiments and theory reveal how *Polistes* wasps achieve specialization and plasticity

- Molecular upregulation of queen phenotype is balanced by colony-scale repression

- This allows *Polistes* to distinguish between intrinsic and extrinsic perturbations

 Patalano et al., 2022, Cell Systems *13*, 768–779
September 21, 2022 © 2022 The Authors. Published by Elsevier Inc.

# Cell Systems

## Report

# Self-organization of plasticity and specialization in a primitively social insect

Solenn Patalano,[1,14,16,*] Adolfo Alsina,[2,14] Carlos Gregorio-Rodríguez,[3] Martin Bachman,[4,5,6] Stephanie Dreier,[7] Irene Hernando-Herraez,[1] Paulin Nana,[8] Shankar Balasubramanian,[5,6,9] Seirian Sumner,[10] Wolf Reik,[1,11,12,15,17,*] and Steffen Rulands[2,13,15,18,*]

[1]Epigenetics Programme, Babraham Institute, Cambridge CB22 3AT, UK
[2]Max Planck Institute for the Physics of Complex Systems, Noethnitzer Str. 38, 01187 Dresden, Germany
[3]Departamento de Sistemas Informáticos y Computación, Universidad Complutense de Madrid, Plaza de Ciencias, 3, 28040 Madrid, Spain
[4]Discovery Science and Technology, Medicines Discovery Catapult, Alderley Park, Cheshire SK10 4GT, UK
[5]Cancer Research UK Cambridge Institute, University of Cambridge, Li Ka Shing Centre, Cambridge CB2 ORE, UK
[6]Yusuf Hamied Department of Chemistry, University of Cambridge, Cambridge CB2 1EW, UK
[7]Institute of Zoology, Zoological Society of London, Regent's Park, London NW1 4RY, UK
[8]Faculty of Agronomy and Agricultural Sciences, School of Wood, Water and Natural Resources, University of Dschang, Ebolowa Campus, P.O. Box 786, Ebolowa, Cameroon
[9]School of Clinical Medicine, University of Cambridge, CB2 0SP, Cambridge, UK
[10]Centre for Biodiversity and Environment Research, Department of Genetics Evolution and Environment, Division of Biosciences, University College London, Gower Street, London WC1E 6BT, UK
[11]Wellcome Trust Sanger Institute, Hinxton CB10 1SA, UK
[12]Centre for Trophoblast Research, University of Cambridge, Cambridge CB2 3EG, UK
[13]Center for Systems Biology Dresden, Pfotenhauer Str. 108, 01307 Dresden, Germany
[14]These authors contributed equally
[15]Senior author
[16]Present address: B.S.R.C "Alexander Fleming," 34 Fleming Street, 16672 Vari, Greece
[17]Present address: Altos Labs Cambridge Institute, Cambridge CB21 6GP, UK
[18]Lead contact
*Correspondence: patalano@fleming.gr (S.P.), wreik@altoslabs.com (W.R.), rulands@pks.mpg.de (S.R.)

## SUMMARY

Biological systems have the capacity to not only build and robustly maintain complex structures but also to rapidly break up and rebuild such structures. Here, using primitive societies of *Polistes* wasps, we show that both robust specialization and rapid plasticity are emergent properties of multi-scale dynamics. We combine theory with experiments that, after perturbing the social structure by removing the queen, correlate time-resolved multi-omics with video recordings. We show that the queen-worker dimorphism relies on the balance between the development of a molecular queen phenotype in all insects and colony-scale inhibition of this phenotype via asymmetric interactions. This allows *Polistes* to be stable against intrinsic perturbations of molecular states while reacting plastically to extrinsic cues affecting the whole society. Long-term stability of the social structure is reinforced by dynamic DNA methylation. Our study provides a general principle of how both specialization and plasticity can be achieved in biological systems. A record of this paper's transparent peer review process is included in the supplemental information.

## INTRODUCTION

Biological systems have the remarkable capacity to build and maintain complex spatio-temporal structures. Such structures are often surprisingly robust in noisy environments, and their formation relies on the integration of regulatory processes on vastly different spatial scales of organization, from the molecular level to tissue or population-level feedback (Cross and Greenside, 2009). Although historically theoretical and experimental research has focused on the processes underlying the formation of complex structures (Ocko et al., 2019; Peleg et al., 2018) such

as the self-organization of cells into complex organs (Davies, 2013), in recent years, it has become clear that biological systems also have the remarkable capacity to break up and rebuild these structures (Merrell and Stanger, 2016; Kennedy et al., 2017). As an example, colonies of social insects rely on the long-term specialization of individuals into distinct castes, such as queen and worker polyphenisms (West-Eberhard, 2003). Although such phenotypes can be stable over years in the face of environmental noise, individuals are nevertheless capable of being rapidly phenotypically reprogrammed: upon receipt of specific cues, they undergo a transient phase in which an

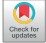

individual's behavior normally associated with a specific caste is rapidly altered in order to perform a different task from the one it performed initially (plasticity) (Herb et al., 2012; Simola et al., 2016; Sheng et al., 2020; Todd et al., 2019).

The regulation of specialization typically relies on the establishment of distinct stable states (bistability) which is often mediated by positive feedback loops (Strogatz, 2015). In bistable systems, transitions between stable phenotypic states require the crossing of a potential or entropic barrier, such that these systems are stable against noise over long periods of time. The same barrier that stabilizes the system, however, mitigates against the rapid re-establishment of a lost phenotype. As in this case, both processes are driven by the same stochastic forces, the time for reprogramming is then comparable with the time of spontaneous phenotype switching (Hänggi et al., 1990) such that stable specialization and rapid plasticity are not simultaneously achievable.

Here, we ask how biological systems can achieve rapid plasticity under specific environmental changes and simultaneously retain stable phenotypic specialization over time. As this problem cannot be resolved using bistability alone, we hypothesized that feedback across different scales of biological organization might give rise to additional degrees of freedom that allow for plasticity while maintaining stable specialization in the steady state.

To test this hypothesis, we used a well-established model system of phenotypic plasticity, colonies of primitively social paper wasps *Polistes canadensis* (Figure 1A) (Ferreira et al., 2013; Patalano et al., 2012, 2015; Sumner et al., 2006; Turillazzi and West-Eberhard, 1996; West-Eberhard, 1969). After the emergence of the foundress' (queen's) daughters (workers), a stable colony of paper wasps is established, with a single reproductive queen and 8–30 non-reproductive workers (specialization). If the queen dies (or is experimentally removed), the remaining workers can rapidly reprogram to generate a unique new queen, hence displaying strong phenotypic plasticity (Miyano, 1986; Strassmann et al., 2004; West-Eberhard, 1969). In order to understand the multi-scale processes regulating specialization and plasticity, we aimed to develop a unique experimental approach that allows us to describe the processes governing the regulation of specialization and plasticity upon *Polistes* queen removal at the molecular and colony scales and the interplay between both. A unique feature of this approach is that it allows measurements to be correlated at the level of the individual across different spatial scales. Based on these experiments, we sought to define a biophysical model that describes these processes and a theory that allows us to understand specialization and plasticity as an emergent property thereof.

We derive the main insight of this work from the synthesis of our experimental and theoretical approaches: we show that *Polistes* integrates processes on different layers of biological organization to distinguish between intrinsic perturbations of molecular states while reacting plastically to extrinsic cues affecting the society as a whole and thereby to simultaneously achieve rapid plasticity and robust specialization. Specifically, the society undergoes a saddle-node bifurcation governed by the population structure itself, thereby simultaneously achieving bistability in the steady state and transient, rapid convergence to the queen phenotype after colony-level perturbations.

## RESULTS

### Upregulation of queen genes in all workers after queen removal

To empirically study the interplay between processes on different scales of biological organization, we took the following experimental approach. After removal of the queen in stable and established colonies, we followed the relaxation dynamics until the re-establishment of a steady state in each colony simultaneously on vastly different spatial scales (Figure 1B): on the colony level by video recordings (individual interactions), on the individual level by characterizing their reproductive states (ovary dissection), and on the molecular level by whole genome multi-omics of the brains of individual insects (RNA sequencing [RNA-seq] of the transcriptome and bisulfite sequencing of the methylome). Although these experiments have individually been conducted in related systems in the past (West-Eberhard, 1969; Theraulaz et al., 1995; Taylor et al., 2021), we, here for the first time, correlate them on the level of individual insects. To characterize the reproductive and molecular dynamics occurring during the reprogramming experiments, we collected a subset of nests at five specific time points after queen removal (Figure 1C). Of the 26 nests monitored in their natural environment, queens were removed from 18 nests 2–3 weeks after the first emergence of workers (Table S1). After queen removal, a rapid reprogramming phase begins (eggless phase), at the end of which (6.55 ± SD 1.97 days) at least one reproductive individual is established (Figure S1A) as evidenced by the observation of egg laying behavior. The subsequent phases were determined on the basis of field observations: 1 day after the identification of at least one reproductive individual (D1 phase), 4–5 days after the establishment of a main reproductive individual (D4 phase), and 14 days after complete stabilization of the colonies (D14 phase).

We initially focused our analysis on the gene expression level, regarded as an important caste regulatory process in *Polistes* (Berens et al., 2015b, 2015a; Patalano et al., 2015; Sumner et al., 2006; Taylor et al., 2021). To identify changes in gene expression associated with queen replacement, we compared brain transcriptomes of all workers from 2 unmanipulated control nests (27 individuals) with transcriptomes of individuals from 3 other nests in which a main reproductive individual had been recently established (24 individuals, D4 phase). 227 genes were differentially expressed between control and D4 nests, with strong enrichment of genes previously associated with reproductive (queen-) phenotypes ($p = 3.1e{-}15$), henceforth referred to as "queen genes" (Patalano et al., 2015; Sumner et al., 2006) (Figure 1D). Consistent with this finding, we also found a strong enrichment of genes that correlate with ovary growth in control nests ($p = 1.5e{-}70$) as well as functional enrichments linked to metabolism, catalysis, and transport of lipids (Figure S1B). For instance, *Vitellogenin* and *Apolipophorin-3*, two genes involved in lipid transport, were strongly upregulated across individuals collected at D4, a conserved pathway activated for reproductive caste division in response to pheromone changes (Figure S1C) (Corona et al., 2016; Hunt et al., 2010; Holman et al., 2019). A principal component analysis confirmed that the workers collected during the D4 phase are all characterized by a shift in their transcriptome profile toward the transcriptome profile of established queens collected during the control phase

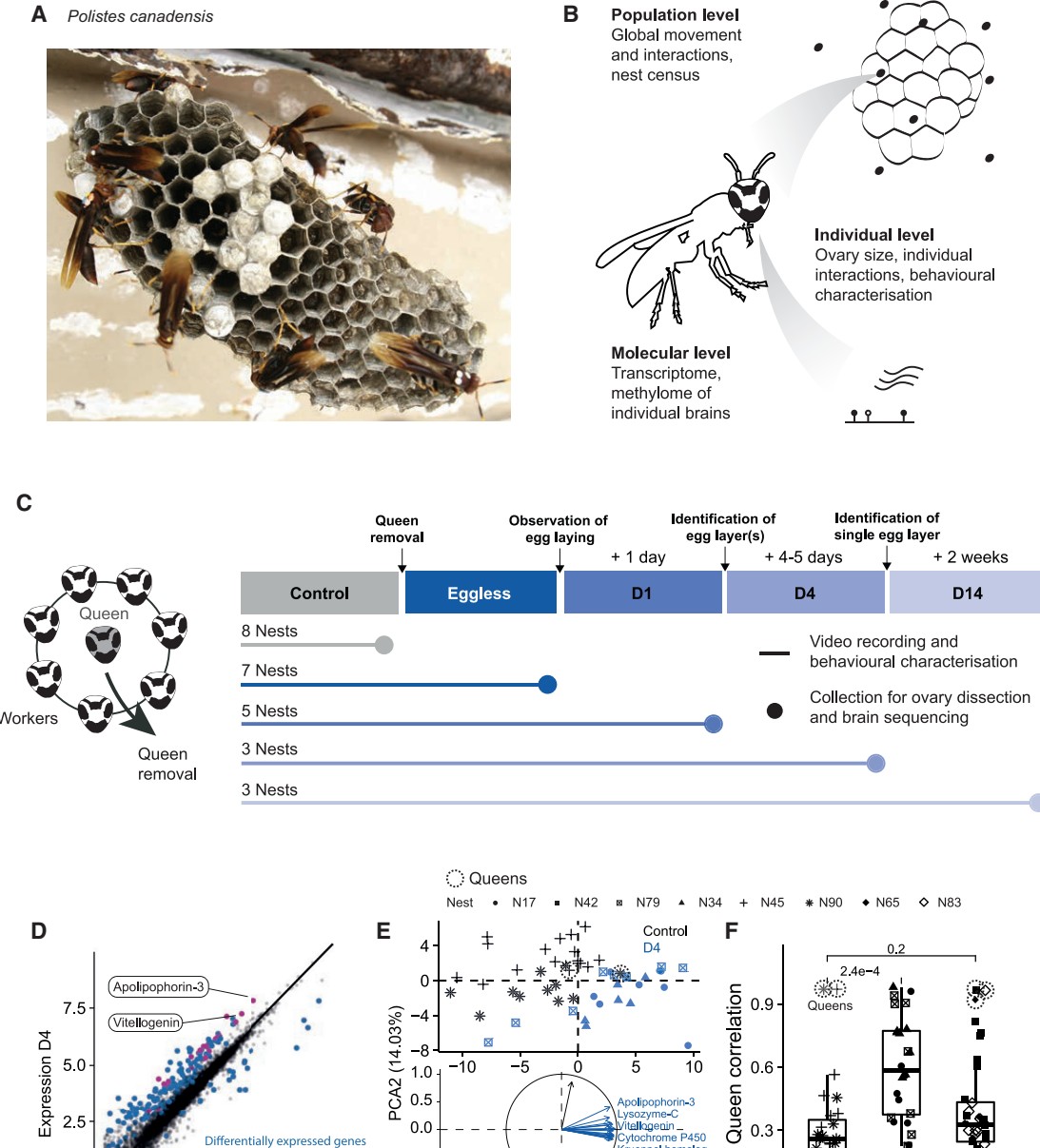

**Figure 1. Upregulation of queen genes during reprogramming**

(A) Photo of a typical nest of *Polistes canadensis*.

(B) Schematic of the multi-scale experimental approach.

(C) Schematic of the experimental time course of queen removal experiments. Control phase: 2–3 weeks prior to queen removal; eggless phase: time between queen removal and egg laying; D1 phase: day or day + 1 after identification of an egg layer(s); D4 phase: day + 4 or +5 after identification of the main egg layer; and D14 phase: 2 weeks after stable egg laying by the new queen.

(D) Gene expression levels in reads per kilobase of transcript per million mapped reads (RPKM), averaged over all individuals collected in control or D4 phases. Differentially expressed genes (DEGs) between phases are marked in blue. DEGs overlapping with queen genes identified in Patalano et al., 2015 are marked in pink.

(E) Principal component analysis of individual insects using DEGs. The radar plot shows the PCs of the 20 most contributing genes (blue: PC1; black: PC2).

(F) Global gene expression correlation of individuals compared with control (established) queens. Center line corresponds to the median, and lower and upper hinges to the 25th and 75th percentiles, respectively. p values were calculated using t tests between control and the two experimental phases and corrected for multiple testing using the Benjamini-Hochberg (BH) method. Control, n = 27; D1, n = 24; D4, n = 36.

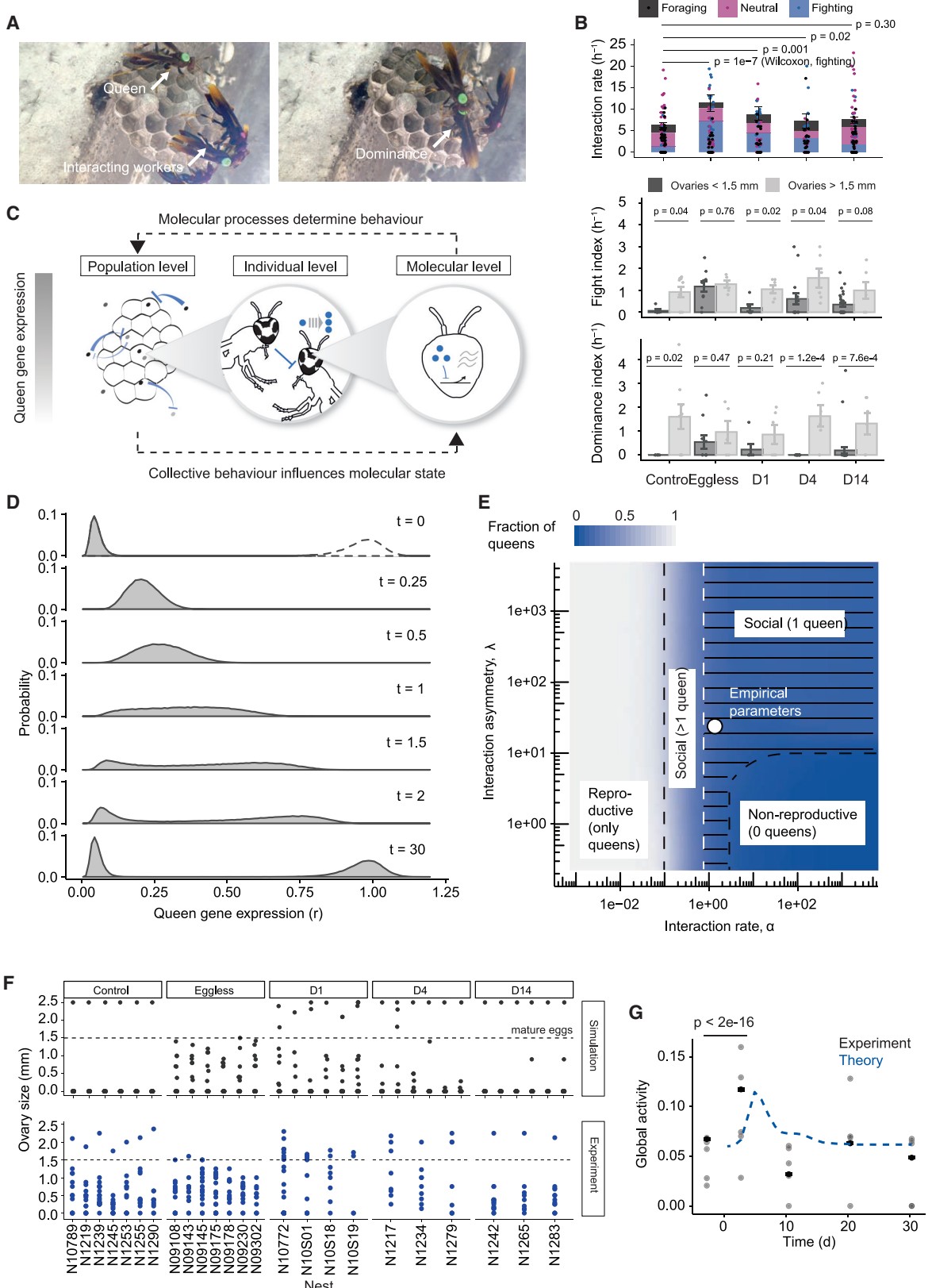

(legend on next page)

(Figure 1E). This observation is further supported by an increase in the correlation of queen gene expression profiles of workers in the D4 phase with queens in control nests (Figure 1F), consistent with the coordinated change of queen gene expression observed after queen removal in another *Polistes* species (Taylor et al., 2021). Taken together, these observations show that at the transcriptome level, queen removal rapidly causes a phenotypic switch to queens in all workers. However, this loss of worker phenotypes after queen removal is transient as individuals collected from 3 nests having already recovered their steady state (34 individuals, D14 phase) show gene expression profiles similar to unmanipulated workers (Figure 1F).

Understanding how the interplay between different scales of biological organization can give rise to specialization and plasticity requires understanding how perturbations on all scales are propagated across the different layers of biological organization. As this is experimentally unfeasible, our strategy is to derive a theory that predicts the multi-scale response of the society to perturbations. Such a theory would constitute a complete biophysical understanding of the multi-scale nest dynamics. As a first step, following the approach of statistical physics, we sought to define the simplest model compatible with the experimental data. In particular, in order for this model to be unsensitive to the partially unknown details of the complex processes underlying the regulation of the society, we did not assume any non-linear terms unless motivated by experimental observations. To begin deriving a biophysical description of the nest dynamics, we describe the molecular state of insect $i$ by a single degree of freedom, $r_i$. This is supported by the observation that different kinds of molecular regulation respond to queen removal on similar time scales (hours), by our observation that the expression levels of different queen genes are correlated and by earlier observations in which *Vitellogenin* and *Apolipophorin* were upregulated within a few hours under hormonal influences resulting from the queen's absence or through induction (Edinger et al., 1997; Hamilton et al., 2016; Röseler, 1977; Röseler and Röseler, 1978). Importantly, our results do not depend on specific assumptions on the dynamics of particular molecular processes. Although $r_i$ therefore represents a coarse-grained description of complex molecular states, we will, for specificity, henceforth refer to $r_i$ as the concentration of queen gene products as an important instance of such states. The time evolution of the probability that a randomly chosen insect $i$ has a concentration of queen gene products $r_i$, $P(r_i, t)$, is given by changes (so called probability fluxes) due to molecular

processes, $J_{mol}$, and feedback by the colony level, $J_{col}$, $\partial_t P(r_i, t) = J_{mol} + J_{col}$. Our RNA-seq experiment shows that in the absence of the queen, individuals constitutively express queen genes (cf. Figures 1D–1F) (Alberts et al., 2015; Hamilton et al., 2016). The flux from the molecular level alone, $J_{mol}$, therefore reads in its simplest mathematical form $J_{mol} = \mathbb{E}^{-1} P(r_i, t) + \mathbb{E}^1 r_i P(r_i, t)$, where the two terms can be interpreted as the production and degradation of queen gene products, respectively, and we have adopted the notation of step operators (van Kampen, 2007), $\mathbb{E}_k^n f(\{x_1, ..., x_N\}) = f(\{x_1, ..., x_k + n, ..., x_N\}) - f(\{x_1, ..., x_N\})$. Time is measured in units of the degradation time of queen gene products, and concentrations are scaled by the steady-state level of expression. Therefore, in the hypothetical case, where the society is regulated on the molecular level alone, the time evolution of $\partial_t P(r_i, t) = J_{mol}$ would lead to a steady state, termed attractor, where all individuals express queen genes at a high level given by a balance between production and degradation ($r_i = 1$). The existence of such a steady state in the molecular dynamics is sufficient for the derivation of all results presented below. As the evolution of dynamical systems is characterized by their attractors, taking into account other molecular processes (hormones, pheromones, gene expression,…) would not alter the conclusions drawn from this model (Strogatz, 2015).

### Increase of dominant behavioral interactions during reprogramming

However, this symmetric convergence of all worker transcriptomes toward queen signatures (Figure 1E) cannot explain the eventual emergence of a single new reproductive individual. Thus, the restoration of a stable state necessarily requires the breaking of this symmetry by a collective process on the colony level.

To study whether there is a colony-level component in the regulation of reprogramming and phenotypic specialization, we analyzed over 17 h of video recordings from 5 nests that had undergone queen removal up to phase D4 (2 nests, 4 phases, 47 ± SD 22 min) and D14 (3 nests, 5 phases, 46 ± SD 21 min). We used computer vision analyses to quantify global activity (overall movement at the nest level, Figures 2A and 2B; Video S1) and manual video inspection to classify and quantify 6 stereotypic individual interactions (Figure 2A). We found a significant increase in both nest activity and the rate of interactions (Figures 2B, S2C, and S2D). Particularly, the rate of aggressive behaviors across individuals during the reprogramming process, especially during the eggless phase, and until the D4 phase increased

---

**Figure 2. Plasticity and specialization emerge from antagonistic interactions on the molecular and the colony scales**

(A) Snapshots of videos showing the interaction between wasps.

(B) Quantification of the interaction rate separated by type (top), the fight index (fraction of fighting interactions among all interactions per individual, middle), and the dominance index (fraction of dominant interactions among fighting interactions per individual, bottom). Bars depict mean ± SEM and dots represent individuals. p values were calculated from a Wilcoxon signed rank test between the corresponding phases and corrected for multiple testing using the BH method (n = 5 nests, 80 individuals).

(C) Schematic of the model (more details in supplemental theory).

(D) Probability densities of queen gene expression levels in a nest after queen removal obtained by stochastic simulations.

(E) Phase diagram showing different types of social structures as a function of the interaction asymmetry ($\lambda$) and the interaction rate ($\alpha$). Empirical parameter values are marked by a white circle (STAR Methods).

(F) Experimental measurements of the size of the most mature egg of individual wasps for different nests across the whole duration of the reprogramming process (top) (n = 385 individuals). Exemplary stochastic trajectories (bottom).

(G) Prediction of global activity changes by the model. A t test was performed on the null hypothesis that average activity is equal in control and reprogrammed nests.

significantly, suggesting that such interactions could contribute to the re-establishment of the steady state.

To quantify a potential asymmetry in interactions, we calculated, for each individual, indices representing the frequency of their involvement in aggressive interactions (fight index) and the probability of dominance during these altercations (dominance index). In order to associate these colony-scale indices with those obtained at the molecular scale, we used the reproductive characteristics of each individual by measuring egg development. After classifying the individuals according to their ovary size, the analysis of fight and dominance indices revealed that in control nests, there is a significant asymmetry of interactions between individuals where only individuals with evidence of developed ovaries—typically the queens—are involved in fights and successfully dominate nestmates (Figure 2B). On the other hand, after the removal of the queen, this asymmetry broke down from the eggless phase, where all individuals, regardless of the size of their ovaries, start to interact in an aggressive manner, until phase D4 (Figure 2B). These observations are consistent with the collective upregulation of their queen genes as observed earlier (STAR Methods section; Figure S2E). After D14, the asymmetry in interactions is re-established. Taken together, these experimental results show that the development of ovaries positively correlates with the rate of interactions (fighting index) and with the probability of behavioral dominance (dominance index) in a given interaction event (Hamilton et al., 2016; Turillazzi and West-Eberhard, 1996). Because queen gene expression correlates with ovary development (Figure S2E), this indicates a parallel relationship between queen gene expression and interactions.

Based on these findings, in the simplest mathematical description reflecting these observations, the rate of interactions between two wasps $i$ and $j$ is proportional to their queen gene expression levels, $r_i r_j$. The asymmetry of interactions observed in control nests requires a factor that decreases monotonically with the difference of their queen gene expression, $\sigma(r_i - r_j)$. The function $\sigma$ decreases monotonically on a scale $1/\lambda$, such that $\lambda$ is proportional to the degree by which gene expression levels can be mutually discriminated (supplemental theory). As a result, the rate with which individual $i$ is subject to subdominant interactions with individual $j$ is $K(r_i, r_j) = r_i r_j \sigma(r_i - r_j)$. In order to break the molecular symmetry across workers, fighting interactions must have a repressive effect on the expression of queen genes in the subdominant insect, which is supported by our transcriptome results at D14 (Figure 1F) and by previous experimental observations showing inhibition of endocrine activities in subdominant individuals (Röseler et al., 1983). The biological details of such a repression usually involve the prolonged increase of the concentration of repressor molecules, which in our model reduces the rate of queen gene expression. By integrating out the degrees of freedom describing such repressor molecules (supplemental theory), the time evolution of $P(\{r_1, \ldots, r_N\}, t)$ effectively becomes dependent on its history, which in mathematical terms means that the process becomes non-Markovian. However, in the limit of time scales much longer than typical interaction times, the effect of the colony feedback on the molecular state of individual $i$ can be entirely absorbed in $J_{col}$ which can be approximately written as $J_{col} = \mathbb{E}^m \alpha P(r_i, t) \sum_{j \neq i} K(r_i, r_j)$, where the parameter $\alpha$ is the interaction rate per individual and $m$ denotes the average reduction of queen

gene products as a result of one interaction. A complete derivation of the model is given in the supplemental theory. Taken together, in this biophysical model (Figure 2C), collective dynamics on the colony level affect queen gene expression on the molecular scale and, vice versa, queen gene expression determines the rate and outcomes of interactions between pairs of individuals on the colony scale.

## Antagonistic dynamics on the molecular and colony levels allow the emergence of a single queen

To test the hypothesis that phenotypic specialization and plasticity both result from the combination of upregulation of queen genes and their colony-level repression, we next considered the stochastic dynamics of the entire nest described by the joint probability $P(\{r_1, \ldots, r_N\}, t)$ of finding a given nest composition $\{r_1, \ldots, r_N\}$ at a time $t$. The time evolution of this probability is governed by contributions stemming from all individuals, $\partial_t P(\{r_1, \ldots, r_N\}, t) = \sum_i (J^i_{mol} + J^i_{col})$, with the molecular- and colony-induced fluxes defined as $J^i_{mol} = \mathbb{E}_i^{-1} P(\{r_1, \ldots, r_N\}, t) + \mathbb{E}_i^1 r_i P(\{r_1, \ldots, r_N\}, t)$ and $J^i_{col} = \mathbb{E}_i^m \alpha P(\{r_1, \ldots, r_N\}, t) \sum_{j \neq i} K(r_i, r_j)$, respectively.

Our calculations and stochastic simulations show that starting from a configuration of individuals with low expression levels of queen genes (workers), the transient absence of repressive interactions leads to a global increase in queen gene expression levels. This is a emergent consequence of the model definition, and it is in agreement with the experimental observation of hormonal and gene expression changes in response to queen removal (Edinger et al., 1997; Hamilton et al., 2016; Röseler, 1977; Röseler and Röseler, 1978). Ultimately, the competition between upregulation on the molecular scale and repression on the population level indeed leads to a bimodal distribution of phenotypes in the population, where queens and workers are clearly distinguished by their molecular states (Figure 2D). Although experimentally the outcome is the emergence of a single queen, the model captures a range of potential asymptotic population structures depending on the interaction rate, $\alpha$, and the degree of the sensitivity of dominance behavior, $\lambda$ (asymmetry). More specifically, for weak and symmetric interactions, all wasps attain reproductive phenotypes, whereas for sufficiently frequent and asymmetric interactions, exactly a single queen phenotype is predicted to emerge, without the need for tuning of parameters (Figure 2E).

By design, this model does not aim to describe the full complexity of the biological processes regulating *Polistes* society. However, it nevertheless quantitatively predicts key experimental observations, such as the time evolution of ovary sizes and nest activity during the reprogramming dynamics (Figure S2G) and the non-trivial transient emergence of multiple egg layers just after the reprogramming process (Figures S2F–S2H). In addition to our experimental observations from which we derived the kinetic processes underlying the model, these predictions further support the mechanistic basis of the model.

## Specialization and plasticity are simultaneously achieved by distinguishing between intrinsic and extrinsic perturbations

To gain mechanistic insight into how robust specialization and plasticity are simultaneously achieved in the wasp society, it is

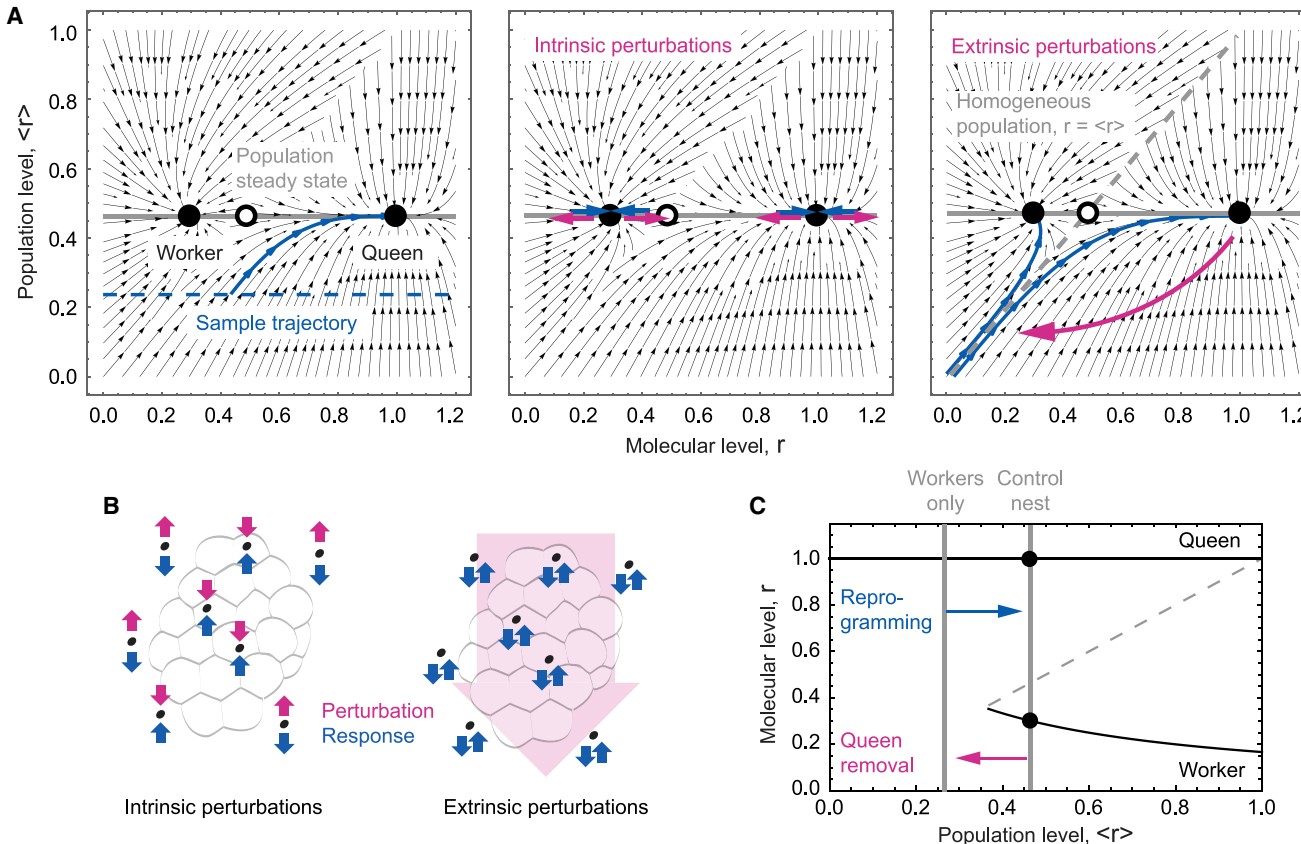

**Figure 3. The colony reacts differently to perturbations across scales**

(A) Phase portrait depicting the joint dynamics of the population structure $f(r,\tau)$, represented by the average $\langle r \rangle$, and an individual insect, represented by its queen gene expression, $r$. Left: unperturbed dynamics; middle: intrinsic perturbations; right: extrinsic perturbations. The gray line represents the steady-state average queen gene expression in the colony, the dashed blue line the initial average queen gene expression of a hypothetical queenless population, and the solid blue line the temporal evolution of the queen gene expression level of a particular individual from this colony. In the left and right panels, pink lines denote perturbation and blue lines the ensuing relaxation dynamics.

(B) Schematic depicting the response to intrinsic and extrinsic perturbations.

(C) Bifurcation diagram. Stable attractors are depicted as solid black lines, and unstable states are represented by the dashed gray line.

instructive to consider the dynamics in a mathematical limit where stochastic fluctuations are negligible (mean-field limit). To derive such a description, we considered the time evolution of a single individual embedded into a nest with a fixed composition $P(\{r_1, ..., r_N\}, t)$ (Hartree approximation) and calculated the conditional probability of finding this individual in a state $r$ for a given nest composition. In this (mean-field) limit, the time evolution of the number of individuals that have dimensionless gene expression level $r$ at dimensionless time $\tau$, $f(r, \tau)$, can be obtained in the limit of time scales much larger than typical interaction times. We find that

$$\partial_\tau f(r,\tau) + \underbrace{\partial_r[(1-r)f(r,\tau)]}_{\text{Molecular scale}} = \tilde{\alpha}\partial_r\underbrace{\left[\int_0^\infty K(r,r')f(r',\tau)\,dr'\right]}_{\text{Population scale}},$$

with interaction kernel $K(r,r') = rr'\sigma(r'-r)$ and rescaled interaction rate $\tilde{\alpha}$ (supplemental theory). This approximation accurately describes the structure of the phase space of the

full stochastic dynamics, as validated by comparison to stochastic simulations (Figures S3A and S3B; supplemental theory). The dynamics reach a steady state if processes on the molecular scale (second term on the left-hand side) are balanced by the effect of colony-level feedback (term on the right-hand side). It is therefore instructive to consider the co-evolution of the molecular dynamics—represented by the expression level of queen genes, $r$, and the population composition, $f(r,\tau)$. To this end, we computed the time evolution of a hypothetical individual with a given expression value of queen genes and the corresponding evolution of the population composition (Figure 3A, left; Video S2). The co-evolution of both scales is represented by black arrows, with an exemplary trajectory highlighted in blue. Our analysis showed that the dynamics relax to a steady-state composition of the population (represented by a specific distribution of queen gene expression values, $f$, with an average value of $r = (\sqrt{2\tilde{\alpha}+1} - 1)/\tilde{\alpha}$). This steady state comprises two attractors of the microscopic dynamics at $r_2 = 1/(1+\tilde{\alpha}r)$ and $r_0 = 1$, corresponding to the worker and queen phenotype, respectively, and giving

rise to a bimodal population composition as observed in our stochastic simulations (Figure 2D).

Specialized phenotypes are subject to different sources of perturbations: intrinsic perturbations, such as gene expression noise, and extrinsic perturbations, such as the removal of the queen. Intrinsic perturbations by definition affect individuals independently such that the population composition remains unaffected. Such a perturbation leads to an opposing response in queen gene expression dynamics, such that this perturbation is actively suppressed by the interplay between molecular and colony-scale dynamics (Figure 3A, middle, horizontal displacements). Extrinsic perturbations affect all individuals in the society in a correlated manner, perturbing both the population composition, $f$, and gene expression levels, $r$. As a result of such perturbations, the system converges to a "plastic" state that separates convergence to the worker and the queen phenotype (separatrix). In this state, individuals have the capacity to evolve toward any of the two attractors and therefore become either worker or a queen (Figure 3A, right, diagonal displacement). Mathematically, the system undergoes a saddle-node bifurcation with the population structure, $f$, acting as a functional bifurcation parameter (Figures 3B and 3C). Taken together, the central result of this study obtained from the synthesis of our multi-scale experimental and theoretical approaches is that *Polistes* integrates antagonistic dynamics on different scales to distinguish between intrinsic, molecular-level perturbations (which are uncorrelated between individuals) and extrinsic, population-level perturbations (which are correlated between individuals), reacting stably to the former ones and plastically to the latter ones. Therefore, the interplay between dynamics on different biological scales of organization allows *Polistes* to simultaneously achieve robust specialization and rapid plasticity.

## DNA methylation may stabilize the social structure against strong fluctuations

The preceding analysis predicts that the steady state is metastable with respect to perturbations that are uncorrelated across insects. The precise time scale of escape from this metastable state depends on the specific rates describing the dynamics on the molecular scale and cannot be rigorously deduced from this mean-field analysis. We will in the following estimate the quantitative value of the time scale of the nest stability considering strong noise. By employing an argument by contradiction, we will show with a simple statistical argument that the empirical time scale of nest stability necessitates processes reducing fluctuations on the molecular level.

A *Polistes* worker is, on average, subject to a subdominant interaction 3.7 ± SEM 1.6 times per day in control nests (Figure S4A). Therefore, interactions occur on a comparable time scale—of the order of hours—to typical gene activation or pheromone production times (Edinger et al., 1997; Hamilton et al., 2016; Röseler, 1977; Röseler and Röseler, 1978). As the ratio, $R$, of these time scales is of order one, such a situation necessarily leads to the frequent chance activation of queen genes in workers due to the stochastic timing of interactions. For example, a statistical estimation yields an expected queen persistence time $T = T_{int}(R + 1/[1 - (1 - e^{-R})^N])$, where $T_{int}$ is the average time between two consecutive subdominant interactions of a worker. This queen persistence time would

equate to less than three days for a nest of 20 insects and if the molecular state of insects is stable for 30 h after a queen interaction ($R = 4.6$) (Figure S4B; supplemental theory). The presence of the queen in *Polistes* societies is, however, very stable over several weeks (Southon et al., 2015), which would imply that molecular states such as queen gene expression would be stable in the absence of queen interactions significantly longer than 30 h (Figure S4C). This simple statistical estimate suggests that additional factors stabilize wasp societies against such fluctuations.

As in mammals, *Polistes* have additional, epigenetic, layers of gene expression regulation, including chemical modifications of the DNA, such as the methylation of cytosines (DNA methylation) (Patalano et al., 2015; Standage et al., 2016; Jeong et al., 2018). To understand whether DNA methylation could play a role in controlling fluctuations in the *Polistes* society, we analyzed the brain methylomes of all individuals from an unmanipulated control nest (8 individuals). Since the methylomes originated from the same individuals whose transcriptome had been sequenced before, we were able to correlate DNA methylation and gene expression on the level of single genes in individual insects. In line with previous work (Jeong et al., 2018; Zemach et al., 2010), we confirmed that DNA methylation in gene bodies is positively correlated with gene expression levels (Figure S4C). Moreover, notably, we found that DNA methylation in gene bodies is associated with a reduction of gene expression noise as evidenced by a decrease of the fraction of significantly variable genes with increasing DNA methylation levels (Figures 4B, S4D, and S4E), as it has previously been reported in other systems (Huh et al., 2013). As a mathematical consequence of such a reduction in gene expression variance, $\sigma^2$, due to DNA methylation, the probability of the chance activation of queen genes decreases stronger than exponentially and therefore necessarily increases the stability of the society by orders of magnitude, $T = T_{int}\{R + 1/[1 - (1 - e^{-1/\sigma})^N]\}$ (Figure 4C; supplemental theory). This relation holds independently of the model defined above. Hence, DNA methylation is predicted to contribute to stabilizing the colony as a whole. To test this, we compared the methylome of *Polistes* with another primitive social wasp species, *Belanogaster*, which has a similar level of social organization but reduced colony stability (Tindo and Dejean, 2000). Using mass spectrometry to measure overall DNA methylation levels in brains of insects collected in stable nests, we indeed observed significantly lower global DNA methylation levels in *Belanogaster*, which is in agreement with a stabilizing role of DNA methylation (Figure 4D).

If DNA methylation has a long-term stabilizing role for *Polistes* colonies, we would expect that rapid reprogramming after queen removal would require partial removal of DNA methylation marks. We therefore compared the methylomes of all individuals originating from a nest collected in D4 phase (8 individuals) with those collected in the control phase (8 individuals). We observed a partial but significant erasure of global methylation levels (Control: 1.94 ± SD 0.84%, D4: 1.23 ± SD 0.13%, unpaired t test, p= 0.0335, Figure 4E). This was evident across all genomic regions and particularly at the level of gene bodies and repetitive elements of the genome (Figure 4F). Consistent with DNA methylation loss, the frequency of genes showing significant expression variability increased after queen removal compared

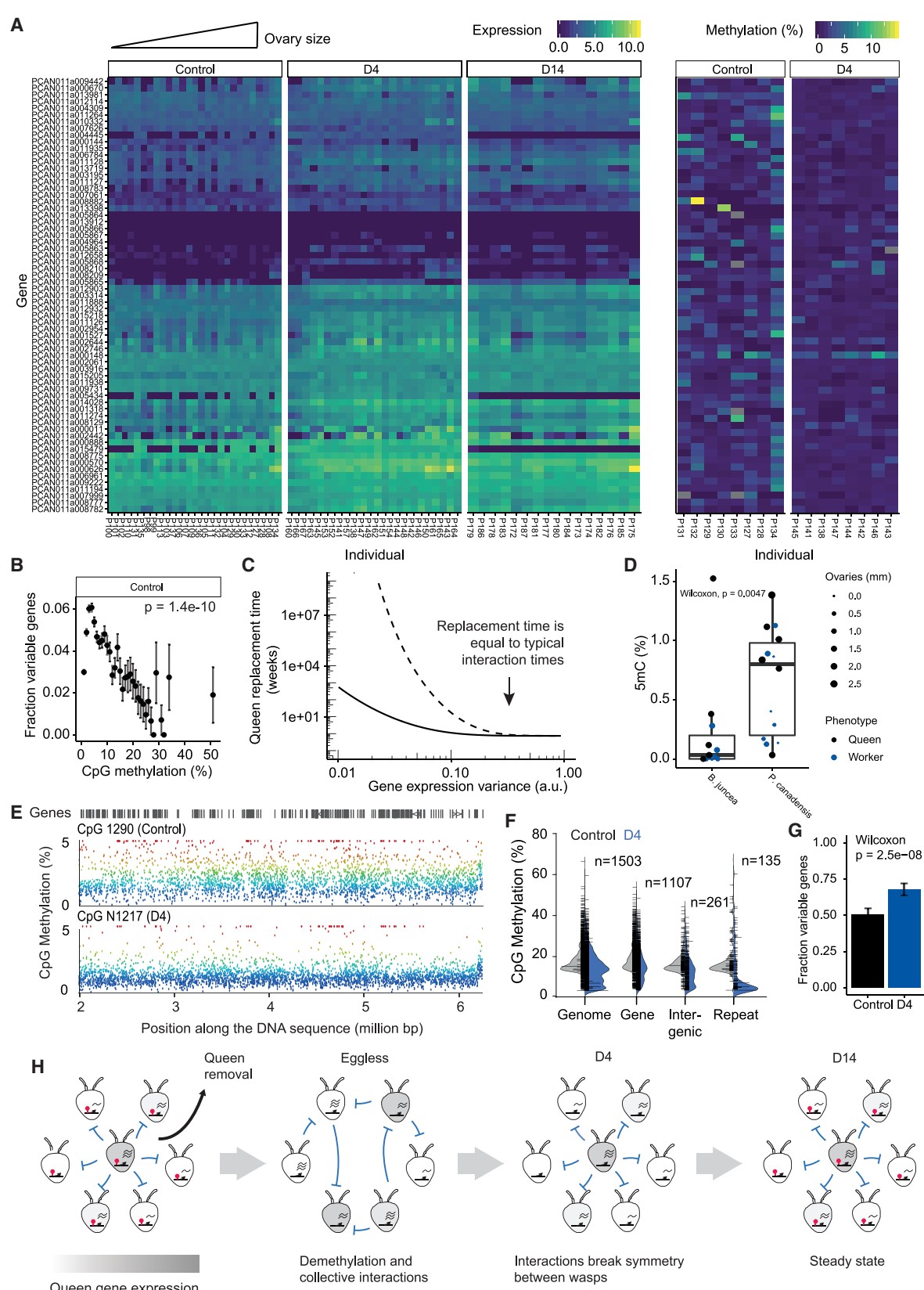

(legend on next page)

with control nests (Figure 4G). To verify that this trend of demethylation was not linked to genetic differences between nests, we validated this result by mass spectrometry by analyzing the global level of DNA methylation of 5 nests from which a subset of individuals was taken before and after queen removal. In each case, we observed a consistently lower rate of DNA methylation in individuals collected during the eggless phase, in which aggressive behaviors are highest (Figure 2B and Figure S5A), confirming the association of DNA methylation levels with the stability of the nest.

Although these observations will require further investigation, in particular for the identification of the mechanisms leading to the depletion of DNA methylation (Figures S5B and S5C) and to establish the reversibility of DNA methylation erasure in *Polistes*, our observations support a role of DNA methylation in stabilizing the *Polistes* society against strong fluctuations.

## DISCUSSION

A combination of experimental and theoretical approaches shows that *Polistes* uses antagonistic dynamics on different spatial scales to distinguish between molecular- and colony-level perturbations, thereby achieving robustness to the former and plasticity to the latter. In our approach, we combined molecular profiling using multi-omics with colony-level video recordings in a way that correlates observations on the molecular, individual, and nest scales to inform a biophysical model and a theory that predicts the propagation of perturbations through multiple scales of organization in *Polistes* societies. In our experiments, we studied colonies of *Polistes* in their natural habitat during field work expeditions in Panama. Although studying a social insect in its natural environment limits the use of molecular perturbation techniques available in laboratories, it allowed us to perform behavioral perturbations under natural environmental fluctuations of *Polistes* nests.

Because of its complexity, a model comprising an accurate description of the biological complexity of the processes regulating the insect society would not allow gaining analytical insight into the response of the society to perturbations. We therefore incorporated the minimal set of assumptions that followed directly from our experimental observations. As the behavior of dynamical systems is governed by the stability of their attractors,

our main conclusions in Figure 3 are robust with respect to the addition of biological complexities going beyond these assumptions. Importantly, in the spirit of statistical physics, variables constituting the model are therefore to be interpreted as coarse-grained descriptions of the complex processes regulating the society. Our theoretical work then allowed us to fully understand the response of the society on all scales to perturbations on the molecular and societal scale and, consequently, the self-organization processes underlying the simultaneous regulation of specialization and plasticity. These results show how societies of primitively social insects can control how fluctuations propagate across scales of biological organization to perform specific functions. Our results also suggest that DNA methylation seems to play an unanticipated role in regulating the stability of the society at the colony level. Our work demonstrates that correlated measurements across scales can give qualitatively new insights into the mechanisms underlying self-organization of biological systems (Figure 4H). Our approach may be more widely applicable to other biological systems of interest and expanded to more complex societal structures (Sasaki et al., 2016). Our work might also help to understand evolutionary processes on much longer time scales (Menzel and Feldmeyer, 2021), as it provides a unified framework for studying the transition from solitary insects to insect societies.

## STAR★METHODS

Detailed methods are provided in the online version of this paper and include the following:

- KEY RESOURCES TABLE
- RESOURCE AVAILABILITY
  - Lead contact
  - Materials availability
  - Data and code availability
- METHOD DETAILS
  - Fieldwork and sample collection
  - Ovaries dissections
  - Video analysis
  - Transcriptome analysis
  - Methylation analysis
  - Biophysical modelling and theory

---

**Figure 4. Epigenetic factors contribute to the stability of the social structure**

(A) Heatmap depicting gene expression (left) and DNA methylation levels (right) of queen genes. Genes are ordered based on hierarchical clustering of gene expression in control nests. Individuals are ordered by their ovary size (small to large, left to right), and queens are marked in bold. Only individuals from oriented libraries are shown.

(B) Fraction of genes showing significant biological variability between workers (see STAR Methods) binned by similar DNA methylation levels in control nests. The p value was determined from a Pearson correlation test of the unbinned data.

(C) Theoretical prediction of queen replacements times as a function of gene expression variance across insects for exponentially (solid line) and normally (dashed line) distributed queen gene expression levels.

(D) Global level of DNA methylation measured by mass spectrometry in *Polistes canadensis* (n = 14 individuals) and *Belonogaster juncea* (n = 11 individuals) control nests. Center line corresponds to the median and lower and upper hinges to the 25th and 75th percentiles, respectively. Only nests with at least 2 individuals collected are shown.

(E) Example of global demethylation. Every dot represents average methylation across individuals in a window containing 50 informative CpGs. Color reflects the methylation level of the probe.

(F) Average of CpG methylation levels for different genomic features across 8 individuals collected before (eggless phase) and after (D4 phase) queen removal. Only windows containing 50 informative CpG with at least 10% average methylation in at least one phase are shown.

(G) Fraction of significantly variable genes (adjusted p-value < 0.05, Wilcoxon) in control and early-commitment phase.

(H) Graphical summary of the sequence of events during the relaxation of the nest.

- ○ Parameter estimation
- QUANTIFICATION AND STATISTICAL ANALYSIS

## SUPPLEMENTAL INFORMATION

## ACKNOWLEDGMENTS

We thank Frank Jülicher, Benjamin D. Simons, and all members of the Reik and Rulands laboratories for helpful discussions. We also thank Nathalie Smerdon at the Wellcome Trust Sanger Institute and Felix Krueger and Simon Andrews at the Babraham Institute for processing sequencing data and bioinformatics support with Illumina sequencing. We thank T. Lengronne, R. Zaurin, R. Southon, and E. Bell for help in the field and all the staff at the Galeta field station and at the Smithsonian Tropical Research Institute Panama for help and logistical support in fieldwork. We also thank P. Vardakas for assistance in the video analyses. This work was conducted under Autoridad Nacional del Ambiente (ANAM) permits #SE/A-33-09, #SE/A-65-10, and #SE/A-20-12 and export permit 10BR004553/DF and 11BR006471/DF. This work was funded by Marie Skłodowska-Curie Individual Fellowship (798082, S.P.), Wellcome Trust (095645/Z/11/Z; W.R.), BBSRC (BB/K010867/1; W.R.), Cancer Research UK (C9681/A18618, C14303/A17197; S.B.), and Wellcome Trust Senior Investigator Award (209441/z/17/z, S.B.), NERC (NE/K011316/1; S.S.). This project has received funding from the European Research Council (ERC) under the European Union's Horizon 2020 research and innovation program (grant agreement no. 950349).

## AUTHOR CONTRIBUTIONS

S.P., A.A., S.S., C.G.-R., W.R., and S.R. designed the research. S.P., P.N, and S.D. conducted field work. S.P. conducted wet lab. M.B. and S.B. acquired mass-spectrometry data. S.P., A.A., C.G.-R., I.H.-H., and S.R. analyzed the data. A.A. and S.R. developed the theory. S.P., A.A., C.G.-R., and S.R. developed code for data analysis and numerical simulations. S.P., A.A., W.R., and S.R. wrote the manuscript.

## DECLARATION OF INTERESTS

W.R. and S.B. are consultants and shareholders of Cambridge Epigenetix.

## SUPPORTING CITATIONS

The following references appear in the supplemental information: Assaf, and Meerson, (2017); Oi et al., (2019); Gardiner, (2004); Bauer et al., (2017); Jörg et al., (2021); Rulands et al., (2013); Biró, and Néda, (2018); and Gillespie, (1977).

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

CelPress

Cell Systems
Report

# STAR★METHODS

## KEY RESOURCES TABLE

| REAGENT or RESOURCE | SOURCE | IDENTIFIER |
|---|---|---|
| **Biological samples** | | |
| *Polistes canadensis* wasps | This paper | N/A |
| **Chemicals, peptides, and recombinant proteins** | | |
| RNA Later | Ambion | R0901 |
| **Critical commercial assays** | | |
| AllPrep DNA/RNA Mini kit | QUIAGEN | 80204 |
| NEB Next Kit | New England Biolabs | 7103 |
| Imprint DNA Modification Kit | Sigma–Aldrich | MOD50 |
| **Deposited data** | | |
| Raw sequencing data | This paper | GEO: GSE144409 |
| *Polistes canadensis* genome | NCBI | GCF_001313835.1 |
| **Software and algorithms** | | |
| Trim Galore v0.4.1 | | www.bioinformatics.babraham.ac.uk/projects/trim_galore/ |
| TopHat v2.0.12 | | https://ccb.jhu.edu/software/tophat/index.shtml |
| Bismark | | https://www.bioinformatics.babraham.ac.uk/projects/bismark/ |
| SeqMonk Version 1.39.0 | | https://www.bioinformatics.babraham.ac.uk/projects/seqmonk/ |
| RepeatMasker v4.0.1 | | https://www.repeatmasker.org |
| R-4.0.2 | | https://www.r-project.org |
| Open CV | | https://opencv.org |
| Custom kinetic Monte-Carlo code | | https://doi.org/10.5281/zenodo.6582146 |

## RESOURCE AVAILABILITY

### Lead contact
Further information and requests for resources and reagents should be directed to and will be fulfilled by the lead contact, Steffen Rulands (rulands@pks.mpg.de).

### Materials availability
This study did not generate new unique reagents.

### Data and code availability
- Sequencing data and their respective NCBI SRA accession numbers can be found in Table S2.
- Raw data on the size of the ovaries (DataSet_ OvariesDissection.xlsx), measurements of methylation by mass spectrometry (DataSet_ MassSpectrometry.xlsx), and both global (DataSet_GlobalNestActivity.xlsx) and individual (DataSet_IndividualBehaviour.xlsx) behaviour activities are available in the supplementary datasets.
- All original code has been deposited at Github (https://doi.org/10.5281/zenodo.6582146) and is publicly available as of the date of publication. The link is listed in the key resources table.
- Any additional information required to reanalyze the data reported in this paper is available from the lead contact upon request.

## METHOD DETAILS

### Fieldwork and sample collection
#### *Polistes canadensis*
Field experiments were conducted between June and August during 3 expeditions to Panama between 2009 and 2012 in the protected area of Punta Galeta, Colon (9°21'30.877''N, 79°54'0.0053''W), under the field collection permits #SE/A-33-09, #SE/A-65-10 and #SE/A-20-12 from the Autoridad Nacional del Ambiente (ANAM) of Panama. Nests were all selected at pre-emergence stages, when only the queen and few co-foundresses were present. Few co-foundresses were present and marked when we initiated the

experiment and none of them had developed mature ovaries during the experiment. All nests were monitored daily in order to mark every new emergent worker and censused every other evening to know the entire nest population. All emergent workers of the *Polistes canadensis* species are mated in early life and then have the potential to become a queen (Turillazzi and West-Eberhard, 1996). The queen was identified by removing an egg and observing who replaced it, as well as from behavioural observations and census data. 26 nests were monitored for three to nine weeks and female wasps representing the different stages of the queen succession were collected. 8 nests were collected unmanipulated (Control phase). Phenotypic reprogramming was induced by removing the queen and an egg in 18 nests and monitored every day to detect any new egg layers. 7 nests were collected before an egglayer appeared (Eggless phase). As soon as a new egg appeared the eggless phase was considered to be over. The identity of this new egg layer was determined over the following day by behavioural observations and then 5 nests were collected (D1 phase). A further 3 nests were monitored and collected 4 to 5 days after the first egg laying occurred (D4 phase). Finally, 3 nests were monitored until 2 weeks after egg laying (D14 phase). All collected nests had similar stage of development (35.6 ±14 days post-emergence) and comparable size (15.6 ± 7.2 individuals) and no sign of parasites or disease. Wasps were collected directly off their nests individually with forceps during the active hours of the day. Their heads were cut off and immediately placed in RNA later solution (Ambion) incubated at 4C overnight to ensure that solution penetrates the brain and kept at -20°C until the dissection of brains. Their bodies were stored in 80% ethanol and kept at -20°C until dissection of ovaries.

### Belonogaster juncea

Three nests of *B. juncea* and their 11 individuals were collected in March 2013 in Ebolowa, Cameroon (2°55′N 11°9′E). Collection, storage and reproductive state assessment of all individuals were done similarly than for *P. canadensis*.

### Ovaries dissections

Reproductive state was assessed for 385 wasps by measuring the first and biggest egg at the entrance of the oviduct. A mature egg has a size of 1.5 mm, which corresponds to the smallest egg size observed and associated with egg-laying. Ovary size was used a proxy for queen gene expression (*vitellogenin* and *apolipophorin-3*) (Röseler et al., 1983; Giray et al., 2005; Röseler, 1977).

### Video analysis

An HD video camera was placed in front of 5 nests during the 2012 fieldwork and video recording was made for at least 30 min during active hours for each phase of the queen removal experiment. Two types of analyses were performed on the videos: quantification of the overall nest activity using computer vision and a classification of wasp behaviours. We use automated machine learning approach to quantify the global movement analysis of the overall nest. Camera movements and unusual captures that might disrupt the quantification were removed. Computer vision analyses (Video S1) were performed by the quantification of the global pixel changes measured between each frame using Python programming language and Open CV (Open Source Computer Vision Library) and normalized by the number of wasps' present. To allow phase and nest inter-comparisons, a normalization of a nest distance with the camera was computed by measuring the diameter of the label used to mark the wasps.

For the individual behavioural analyses, we monitored a total of 400 interactions from 80 individuals and manually classified them into 6 different types of interactions as previously described in *Polistes* (Pratte et al., 1990; Strassmann et al., 2004). The classification was performed manually, reducing the possibility of mis-classification compared to computational approaches. A repetition of the analysis produces results deviating by only 15% from the original analysis. Briefly, DOM: domination over a nestmate (bite, sting, chew, nip or chase). SUB: subordination (in response to DOM or escape). DEC: donor of foraging material to a nestmate. REC: receiver of the DEC. TR: involved in trophallactic exchanges (i.e., exchanges of fluid from one wasp to another). ANT: Involved in antennation. We categorised these types as follows: Fighting as DOM and SUB, foraging as REC and DEC, and neutral interactions as ANT and TR. From this table, Interaction rate (DOM+SUB+DEC+REC+ANT+TR per hour per wasp), Fight index (DOM + SUB / All interactions) and DOM index (DOM/(DOM+SUB)) were calculated for each individual wasp and normalised by the amount of time each wasp was observed on the nest. Wasps present in the nest but not performing any interaction were also included in our analysis. To calculate the rate of subdominant interactions per worker and per hour, i.e., a passive measure of interactions, we considered individuals who were not present in the nest during the video in order to obtain an unbiased measure of this rate. In this specific case, we therefore normalised by the total number of individuals detected in a night-time population census conducted before the video recording.

### Transcriptome analysis

Three phases were analysed by RNA-seq (Control n=2 nests, D4 n=3 nests and D14 n=3 nests). The time points were selected in this way to ensure a representation of different stages of the reprogramming process (control: 0 day, D4: ∼11 days, D14: ∼21 days) and can thus best support the mathematical model. Total RNA was extracted from 87 single brains, using the QIAGENAll PrepDNA/RNAMini Kit according to the manufacturer's instructions. 50 to 200 ng of total RNA was enriched for mRNA using Dynabeads Oligo(dT)25 from Invitrogen in two subsequent steps of purification with fresh beads. Fragmentation was done by incubation of mRNAs for 5 min at 94 °C in the First-Strand Buffer (Invitrogen), and directly followed by cDNA synthesis using a SuperScript III Kit (Invitrogen) according to the manufacturer's instructions. dUTPs were incorporated for second-strand synthesis for library orientation except for individuals originating from nests N1265 and N1283. cDNA was end-repaired, A-tailed, and ligated using the NEB Next Kit (New England Biolabs) according to the manufacturer's instructions. dUTP excision was done before amplification using USER mix (New England Biolabs). Libraries were amplified with 16 cycles using 2× Phusion HF buffer (New England Biolabs). Size selection and

cleaning between steps were performed with the AMPure XP system (Agencourt) to select DNA fragments between 250 bp and 500 bp. Paired-end libraries were sequenced on HiSeq 2500 Illumina platform.

RNA-seq libraries were sequenced on the Illumina HiSeq platform using the default RTA analysis software. RNA-Seq data were trimmed with Trim Galore (v0.4.1, default parameters) and mapped to the *Polistes canadensis* genome assembly GCF_001313835.1 using TopHat v2.0.12 as previously described in (Patalano et al., 2015). Strand specific quantification was performed using RNA-seq pipeline in Seqmonk software Version 1.39.0 (www.bioinformatics.babraham.ac.uk/projects/seqmonk/). To normalize across nests size factors were calculated using the DEseq2 package in R and log transformed with only orientated samples. The list of "queen genes" corresponds to queen-biased genes identified in (Patalano et al., 2015). To identify differentially expressed genes between the control and D4 phase we generated three sets of genes using the DESeq2, EdgeR and intensity difference filters with standard parameters in Seqmonk. P-values were corrected for multiple testing using the Benjamini-Hochberg method and genes were considered differentially expressed if p<0.05. Gene set enrichment analysis was performed using a hypergeometric test. To establish the correlation between gene expression and ovary growth we calculated Pearson's correlation coefficient for each gene across all individuals in a given phase. Genes were considered significantly correlated if p<0.05 after Benjamini-Hochberg correction for multiple testing.

PCA analysis was performed using the FactoMineR package in R. Genes contributing the most to a given PCA dimension were identified from their cos2. To calculate the correlation coefficients to the queen transcriptomic profile, we computed Pearson's correlation coefficients of gene expression for list of genes associated with the queen phenotype (Patalano et al., 2015) between each individual and the two queens in control nests. Shown in Figure 1F is the average of these two Pearson's correlation coefficients for each individual. To calculate biological variance, after removing control queens from the dataset we fitted a technical noise model using the trendVar function of the scran package (version 1.12.1) and computed noise contributions using scran's decomposeVar function. Genes with p-values smaller than 0.1 after multiple testing correction using the Benjamini-Hochberg method were considered significantly variable.

### Methylation analysis
#### Bisulfite sequencing
Two phases were analysed by BS-seq (Control n=1 nest, D4 n=1 nest).

Genomic DNA was extracted from 16 single brains, using the QIAGEN All Prep DNA/RNA Mini Kit according to the manufacturer's instructions. Between 200 ng and 500 ng of input genomic DNA was used per library and spiked with unmethylated lambda DNA to provide an estimation of BS conversion efficiency. DNA was end-repaired, A-tailed, and ligated with a methylation Adaptor Oligo Kit (Illumina) using the NEB Next kit according to the manufacturer's instructions. The adaptor-ligated DNA was treated with sodium-BS using an Imprint DNA Modification Kit from Sigma–Aldrich according to the manufacturer's instructions for the two-step protocol. BS-treated DNA was amplified using KAPA HiFi Uracil + DNA Polymerase (KAPA Biosystems) with 15 cycles. Size selection and cleaning between steps were performed with an AMPure XP system (Agencourt) to select DNA fragments between 250 bp and 500 bp and sequenced at the Sanger Institute using the HiSeq 2500 Illumina platform.

BS-seq libraries were sequenced on the Illumina HiSeq platform using the default RTA analysis software. Raw sequences were trimmed to remove both poor-quality calls and adapters using Trim Galore (version 0.3.5 with default parameters, www.bioinformatics.babraham.ac.uk/projects/trim_galore/). The remaining sequences were then aligned to genome assembly EVM/PASA (Patalano et al., 2015); using Bismark (version 0.12.2, with the parameters: –bowtie2 –score_min L,0,−0.4) and Genome_build: *Polistes canadensis* GCF_001313835.1 from. Quantification was done in SeqMonk over probes which contain 50 CpGs each with a minimum read count of 4. Only 30pb to 1kbp probes were kept. Intergenic and gene features analysis was done using probes that specifically overlap these features. Repetitive genomic regions were identified using RepeatMasker v4.0.1, using the 20120418 repeat libraries limited to Apocrita species. Methylation over a given feature was calculated by averaging the methylation levels of CpGs across the probes overlapping a given feature. Binomial filter from Seqmonk was applied to identify methylated probes having statistically greater variations than the ones observed globally (using CpGs covered by a minimum of 4 reads). Significance was considered only when probes significantly deviated from the global average by at least 10% after Benjamini-Hochberg correction. GO enrichment analysis was performed with OmicsBox using Fisher's Exact test (p<0.01) and reduced to the most specific terms.

#### Mass-Spectrometry
We used this technique to analyse the overall methylation levels in order to: 1) Validate the intra nest demethylation process (n = 5 nests), 2) Compare the methylation levels between the species *P. canadensis* (n = 6 nests, 14 individuals) and *B. juncea* (n = 3 nests, 11 individuals). Genomic DNA were isolated from all single brains of both species using protocol from (Patalano et al., 2015). For the detection of the methyl groups, five hundred nanograms of genomic DNA was incubated with 5 units of DNA Degradase Plus (Zymo Research) at 37 °C for 3 h. The resulting mixture of 2-deoxy- nucleosides was analysed on a Triple Quad 6500 mass spectrometer (AB Sciex) fitted with an Infinity 1290 LC system (Agilent) and an Acquity UPLC HSS T3 column (Waters), using a gradient of water and acetonitrile with 0.1% formic acid. External calibration was performed using synthetic standards, and for accurate quantification, all samples and standards were spiked with isotopically labelled nucleosides (Bachman et al., 2014).

## Biophysical modelling and theory

A detailed and rigorous derivation of the theoretical approach is given in the supplemental theory. Briefly, we defined the minimal stochastic model capable of describing the experimental phenomenology. The evolution of the distribution $P(\{n_i, q_i\}, t)$ of the number of queen gene products, $n$, and the state of queen gene repression, $q$, in each individual at a given time $t$, is given by

$$\frac{d}{dt}P(\{n_i, q_i\}, t) = \sum_{i=1}^{N+1} \left\{ \mu(1-q_i)[P(\{n_i-1, q_i\}, t) - P(\{n_i, q_i\}, t)] + \delta[(n_i+1)P(\{n_i, q_i\}, t) - n_i P(\{n_i, q_i\}, t)] \right.$$
$$\left. + \Gamma(t_i^{int})(1-2q_i)P(\{n_i, 1\}, t) + \omega(2q_i-1)\sum_{j\neq i} K_{ij}P(\{n_i, 0\}, t) \right\},$$

where the first two lines account for the production and degradation of queen gene products and the last line contains the terms describing the effect of the population level interactions on gene expression. Having derived the full stochastic non-Markovian dynamics of the joint probability $P(\{n_i, q_i\}, t)$, we then proceeded to derive a mean-field description valid in the limits of large time scales and negligible fluctuations that still captures the salient features of the structure of the phase space. This mean-field approximation can be written in integro-differential form as

$$\partial_\tau f(r, \tau) + \partial_r \underbrace{(1-r)f(r, \tau)}_{\text{Molecular scale}} = \tilde{\alpha}\partial_r \underbrace{\left[ \int_0^\infty K(r, r')f(r', \tau)\, dr' \right]}_{\text{Population scale}}.$$

The mean-field approximation was validated a posteriori by comparison of the mean-field predictions of the dynamics with single realizations of the stochastic process defined by the master equation for the joint probability. Stochastic simulations were performed using custom code implementing Gillespie's algorithm (Gillespie, 1977). Finally, we analysed the influence of gene expression noise on population stability by performing model-free calculations. Our statistical calculations provide an expression relating population-wide gene expression variance to queen replacement times. We refer the interested reader to the supplemental information for details.

## Parameter estimation

The interaction kernel $K(r_i, r_j)$ depends on two parameters, the dimensionless interaction rate per individual, $\alpha$, and the interaction asymmetry $\lambda$. To locate the empirical parameter values in the phase diagram we estimated these parameters from the experimental data presented in Figure 2B of the main text. To estimate the interaction rate, we calculated the number of fighting interactions per individual in a time interval *of one day* obtaining an estimate of 3.7 interactions per day. The dimensionless interaction rate, $\alpha$, is obtained by rescaling this quantity as indicated in the supplemental theory. Additionally, we counted the number of subdominant queen interactions to obtain a lower bound for the interaction asymmetry parameter $\lambda$. Out of 17 interactions involving queens in the control and late-commitment phases, we observed 0 interactions where the queen was subdominant. Therefore, the maximum likelihood estimates for the error rate using the beta-distribution as a (conjugate) prior is 1/17. Comparing this result with the analytical calculation of the error rate from the definition of the interaction kernel (supplemental theory) yields a lower bound for the asymmetry parameter $\lambda \approx 24$. We take this lower bound to be the empirical estimate of $\lambda$ as the interaction kernel is not very sensitive to changes in the asymmetry parameter in this range.

## QUANTIFICATION AND STATISTICAL ANALYSIS

Statistical details are given in the figure legends and in Table S3. Statistical details relating to bioinformatics processing of sequencing data is given in the method details section. Non-parametric statistical tests were used when data could not be assumed to be normally distributed. P-values were corrected for multiple testing using the Benjamini-Hochberg method where appropriate. The FDR was controlled at a level of alpha=0.05.

