## [Document S2. Transparent peer review records for Patalano et al · Cell Systems]

Self-organisation of plasticity and specialisation in a primitively social insect

Solenn Patalano, Adolfo Alsina, Carlos Gregorio-Rodriguez, Martin Bachman, Stephanie Dreier, Irene Hernando-Herraez, Paulin Nana, Shankar Balasubramanian, Seirian Sumner, Wolf Reik, Steffen Rulands

Summary

Initial Submission: Received Aug 3, 2021

Scientific editor: Bernadett Gaal, DPhil

First round of review: Number of reviewers: 2
2 confidential, 0 signed
Revision invited Nov 23, 2021
Major changes anticipated
Revision received Apr 10, 2022

Second round of review: Number of reviewers: 1
1 original, 0 new
1 confidential, 0 signed
Accepted Aug 5, 2022

This Transparent Peer Review Record is not systematically proofread, type-set, or edited. Special characters, formatting, and equations may fail to render properly. Standard procedural text within the editor's letters has been deleted for the sake of brevity, but all official correspondence specific to the manuscript has been preserved.

Editorial decision letter with reviewers' comments, first round of review

Dear Steffen,

First of all, please accept our sincere apologies for the delay in the review process. As I had mentioned we had encountered difficulties recruiting reviewers for the paper and needed to replace one late in the process. I'm enclosing the comments that reviewers made on your paper, which I hope you will find useful and constructive. As you'll see, they express interest in the study, but they also have a number of criticisms and suggestions. Based on these comments, it seems premature to proceed with the paper in its current form; however, if it's possible to address the concerns raised with additional analysis, we'd be interested in considering a revised version of the manuscript.

As a matter of principle, I usually only invite a revision when I'm reasonably certain that the authors' work will align with the reviewers' concerns and produce a publishable manuscript. In the case of this manuscript, all of the points that the reviewers raise seem clear, important and addressable in principle, but please let me know if you have any concerns.

As you address the reviewers' concerns, it's important that you and I stay on the same page. I'm always happy to talk, either over email or by phone, if you'd like feedback about whether your efforts are moving the manuscript in a productive direction. Do note that we generally consider papers through only one major round of revision, so the revised manuscript would be either accepted or rejected based on the next round of comments we receive from the reviewers. If you have any questions or concerns, please let me know. More technical information and advice about resubmission can be found below my signature. Please read it carefully, as it can save substantial time and effort later.

I look forward to seeing your revised manuscript.

All the best,

Bernadett

Bernadett Gaal, DPhil
Scientific Editor, Cell Systems

Reviewers' comments:

Reviewer #1: The topic tackled by the authors is exciting and the combination of scales on which the queen succession is considered is impressive. I find the work very interesting and there are potentially profound ecological and evolutionary implications.

I would have liked to see these discussed a bit more in the introduction and discussion. But that may be a minority interest in kin-selection and it's role in eusocial insects.

My main comments are mostly of a technical nature where I believe the arguments could be tightened up.

- I found the statistical analysis difficult to follow. It is important to be more specific and upfront about the role of multiple testing corrections. I was not sure in the main text which p values are corrected and which ones were raw or uncorrected. I would like to see the number of tests that were performed in each case. I also would like to see a discussion of why the FDR correction was changed from 0.05 to 0.1. This is regrettably often done, and sometimes there are good reasons, but without explaining why this choice was made it is hard to judge this with confidence. So more transparency would help a lot, e.g. what would the statistical analysis have shown if FDR was controlled at 0.05 throughout? In general I found the statistical analysis not reproducible to the extent that I would like to be able to.

- I could not find sufficient detail on the statistical analysis of the video analysis. In particular I would like to see an assessment of classification/mis-classification rates. The data in Fig 2b is hard to see and interpret. Are the p-values raw or corrected?

- The theoretical modelling is very well explained in the supplementary information. This is very nicely written up. The discussion in the paper is, however, too terse and not complete. For example, the purely additive Ansatz for the change in probability flux of molecular and colony level processes is much better explained in the SI where the absence of interactions is clear as an assumption. I would like to see the same transparency in the main paper, too.

- The non-Markovian dynamics are also nicely derived in the supplementary information. The discussion in the main paper is cryptic and not helpful. Only a fraction of readers would find "Hartree approximation" helpful, I am sure, whereas the more common terminology (also used in the SI throughout) "meanfield" is clearer and to most people more familiar.

- Supplementary Figure 2g and h strikes me as encouraging regarding the modelling efforts. Why are these results relegated to the supplementary material?

Reviewer #2:

This manuscript presents a model, supported by experimental evidence, for how paper wasp colonies maintain a stable social structure while also being able to rapidly reconfigure in response to external perturbation. Colonies have a stable division between a single reproductive queen and dozens of non-reproductive workers. This study posits that all workers have a constitutive tendency to become queens, but this development is inhibited by queen-initiated dominance interactions. If the queen is removed, all workers begin to develop a queen-like phenotype, resulting in an increase in dominance interactions and a subsequent re-repression of queen development in most workers. Repression is posited to result from a tendency of dominance interactions to suppress queen phenotype in losers of the interactions.

Most of the components of this model seem reasonable. I think many of the basic ideas about self-activation and mutual inhibition have been put forward in earlier work. However, a strength of the study is the combined behavioral, physiological, and molecular evidence for rapid emergence of a queen-like phenotype in large numbers of workers after queen removal.

Another is the development of a credible dynamical model showing how the basic assumptions of the model can lead to the combination of stability and plasticity observed in the experiments. This model shows that perturbations in noisy caste determination mechanisms within single wasps will be damped out in a queenright colony; only if queen removal sharply lowers the overall amount of queen-determining factors will the colony enter a regime where changes in caste become possible. This

combination of dynamic modeling and rich evidence at multiple levels of organization is a novel and promising advance in the study of these societies.

The section on the role of DNA methylation muddles the picture somewhat. After describing how the dynamical system of the colony creates stable worker and queen phenotypes, the paper then gives results contradicting this stability; namely, chance activation of queen genes is said to lead to an expected queen persistence time of only a few days. The authors then go on to present interesting evidence for how DNA methylation might make the social structure stable in the face of such perturbations. However, the paper should be modified to explain how this instability is consistent with the apparently contradictory model results previously described and summarized in Figure 3.

It would also help if the paper was clearer about which aspects of the model have experimental evidence and which do not. For example, I did not see evidence for the key idea that losing a dominance interaction reverses the queen-like molecular, physiological, and behavioral phenotype.

The authors might consider re-writing the abstract to be less generic. I understand the need to communicate the generality of their results, but it is very hard to tell from the abstract what is the actual phenomenon under study (regulating queen number in wasp colonies) or the basic idea of how it works (a dynamic balance between constitutive queen development in all colony members and inhibition of queen development by established queens).

The authors might consider exploring whether their model can account for a greater variety of social structures than the three they show in Figure 2e. For example, can it account for a linear hierarchy or the oligarchies seen in certain ants? They might have a look at the paper by Sasaki et al. (*American Naturalist*, 2016) that presents a very different kind of model of related questions.

Some of the figure legends could use more explication. For example, they could better describe the meaning of the dashed blue and gray lines in Figure 3a and give a quick definition of fight index and dominance index in Figure 2b. In Figure 2e, it's hard to understand why there is a Non-reproductive phase. Does this mean there are no queens? The color coding does not suggest any fewer queens there than in the Social zone. In Figure 1e, why is one of the arrows in the radar plot black?

Authors' response to the reviewers' first round comments

Attached.

Editorial decision letter with reviewers' comments, second round of review

Dear Steffen,

I'm very pleased to let you know that Reviewer 2's comments on your revised manuscript are back,

and while we have not heard from Reviewer 1, we are happy to proceed, and only a few minor, editorially-guided changes are needed to move forward towards publication.

I've made some suggestions about your manuscript within the "Editorial Notes" section, below. Please consider my editorial suggestions carefully, ask any questions of me that you need, make all warranted changes, and then upload your final files into Editorial Manager.

As you look forward to acceptance, please do consider submitting one of the protocols you've developed in this paper to STAR Protocols, or extending this offer to one of your trainees. STAR Protocols is geared towards trainees and its key purpose is to provide complete and consistent instructions for how to conduct reproducible experiments. If you have any questions, please email starprotocols@cell.com.

I'm looking forward to going through these last steps with you. Although we ask that our editorially-guided changes be your primary focus for the moment, you may wish to consult our FAQ (final formatting checks tab) to make the final steps to publication go more smoothly. More technical information can be found below my signature, and please let me know if you have any questions.

All the best,

Bernadett

Bernadett Gaal, DPhil
Editor-in-Chief, Cell Systems

Editorial Notes

Transparent Peer Review: Thank you for electing to make your manuscript's peer review process transparent. As part of our approach to Transparent Peer Review, we ask that you add the following sentence to the end of your abstract: "A record of this paper's Transparent Peer Review process is included in the Supplemental Information." Note that this **doesn't** count towards your 150 word total!

Also, if you've deposited your work on a preprint server, that's great! Please drop me a quick email with your preprint's DOI and I'll make sure it's properly credited within your Transparent Peer Review record.

Manuscript Text:

- House style disallows editorializing within the text (e.g. strikingly, surprisingly, importantly, uniquely etc.), especially the Results section. These terms are a distraction and they aren't needed—your excellent observations are certainly impactful enough to stand on their own. Please remove these words and others like them. "Notably" is suitably neutral to use once or twice if absolutely necessary.

- We don't allow "priority claims" (e.g. new, novel, etc.). For a discussion of why, read: <http://crosstalk.cell.com/blog/getting-priorities-right-with-novelty-claims>, <http://crosstalk.cell.com/blog/novel-insights-into-priority-claims>.

Figures and Legends: Remember that you have 2/4/7 full-page, multi-panel figures! It's best practice not to make your reader hunt for what they need in the supplement, so please consider moving **XXXX** to the main figures, specifically Figure X.

Also, please look over your figures keeping the following in mind: *<insert/delete as necessary>*

- Please ensure that every time you have used a graph, you have defined "n's" specifically and listed statistical tests within your figure legend.
- Please ensure that if you include representative images within your figures, a "representative of XXX individual cells"-type statement is made in the legend.
- Please ensure that all figures and tables included in your point-by-point response to the reviewers' comments are present within the final version of the paper, either within the main text or within the Supplemental Information.

STAR Methods:

If you are using GitHub, please follow the instructions here to archive a "version of record" of your GitHub repo at Zenodo, then report the resulting DOI. Additionally, please note that the Cell Systems strongly recommends that you also include an explicit reference to any scripts you may have used throughout your analysis or to generate your figures within section 2.

Cell Systems has taken the almost political stance that theory papers deserve the same "weight" as experimental papers, and we try very hard to treat both consistently. We believe that the deep mathematical exploration of problems that's usually found in a stand-alone Supplement is roughly equivalent to the Experimental Procedures section of experimental papers. Just as one must understand the methods used to appreciate the strengths and weaknesses of the data those methods generate, understanding the mathematics is necessary for understanding the Results in a deep way. For this reason, we recommend that you consider moving at least some of the Supplemental Theory section into the Model Details section of the STAR Methods, and follow the rest of the STAR Methods instructions as well as you can.

Thank you!

Reviewer comments:

Reviewer #2: The authors have done a thorough job of responding to earlier critiques. The revised manuscript is a strong contribution.

We thank both reviewers for their appreciation of our work and their constructive comments which we believe improved the clarity of the manuscript. All issues raised by the reviewers have been addressed in the revised version of the manuscript. Changes to the text are marked in blue.

Reviewer #1: The topic tackled by the authors is exciting and the combination of scales on which the queen succession is considered is impressive. I find the work very interesting and there are potentially profound ecological and evolutionary implications.

I would have liked to see these discussed a bit more in the introduction and discussion. But that may be a minority interest in kin-selection and its role in eusocial insects.

We thank the reviewer for their appreciation of our work. We agree that this work could potentially be applied in these contexts. We are conscious that a discussion in the introduction might shift the attention of readers away from the primary focus of our work and we therefore only introduce key concepts important for the primary research question in the introduction. We now mention further potential ecological and evolutionary implications of our model in the discussion.

My main comments are mostly of a technical nature where I believe the arguments could be tightened up.

- I found the statistical analysis difficult to follow. It is important to be more specific and upfront about the role of multiple testing corrections. I was not sure in the main text which p values are corrected and which ones were raw or uncorrected. I would like to see the number of tests that were performed in each case.

Only the p-values in Figs. 1f, 2b (middle, bottom), 4d and 4g were not corrected for multiple testing as in these cases only a single or very few comparisons were performed. All other p-values were corrected using the default algorithm used in R's p.adjust function. Contrary to what was written in the methods section we found that this corresponds to the Holm method, not the BH method. We edited the manuscript such that all corrected p-values now correctly reflect the BH method. This change did not affect any of the results.

The table below, which we include as a Supplementary Table in the revised version of the manuscript, lists all p-values mentioned in the manuscript along with the test that was used and the correction for multiple testing if applicable.

Figure	Comparison	p-value	Adjusted p (BH)	Adjusted p (Holm)	Number of tests	Test type
1f	Control-D4	7.85×10^{-5}	2.36×10^{-4}	2.36×10^{-4}	2	t-test
1f	Control-D14	0.20	0.20	0.20	2	t-test
2b(top)	Control-Eggless	3.13×10^{-6}	1.25×10^{-5}	1.25×10^{-5}	4	wilcoxon
2b(top)	Control-D1	0.005	0.01	0.015	4	wilcoxon
2b(top)	Control-D4	0.074	0.0987	0.148	4	wilcoxon
2b(top)	Control-D14	0.72	0.72	0.72	4	wilcoxon
2b(mid)	Queen-workers	0.041	0.0683	0.164	5	wilcoxon

	(Control)					
2b(mid)	Queen-workers (Eggless)	0.759	0.759	0.759	5	wilcoxon
2b(mid)	Queen-workers (D1)	0.018	0.0683	0.09	5	wilcoxon
2b(mid)	Queen-workers (D4)	0.041	0.0683	0.164	5	wilcoxon
2b(mid)	Queen-workers (D14)	0.081	0.101	0.164	5	wilcoxon
2b(bot)	Queen-workers (Control)	0.02	0.033	0.06	5	wilcoxon
2b(bot)	Queen-workers (Eggless)	0.465	0.465	0.465	5	wilcoxon
2b(bot)	Queen-workers (D1)	0.214	0.268	0.428	5	wilcoxon
2b(bot)	Queen-workers (D4)	$1.16 \cdot 10^{-4}$	$5.8 \cdot 10^{-4}$	$5.8 \cdot 10^{-4}$	5	wilcoxon
2b(bot)	Queen-workers (D14)	$7.58 \cdot 10^{-4}$	$1.90 \cdot 10^{-4}$	$3.03 \cdot 10^{-3}$	5	wilcoxon
2g	Control-Eggless	$<2.2 \cdot 10^{-16}$	-	-		
4b		$1.4 \cdot 10^{-10}$	-	-		pearson
4d	B.junceae - P.canadensis	$4.7 \cdot 10^{-3}$	-	-		wilcoxon
4g	Control-D4	$2.5 \cdot 10^{-8}$	-	-		wilcoxon
S1c(l)	Control-D4	$5.24 \cdot 10^{-7}$	-	-		t-test
S1c(r)	Control-D4	$1.75 \cdot 10^{-7}$	-	-		t-test
S2d	Control-Eggless	$<2.2 \cdot 10^{-16}$	-	-		t-test
S2e	Vitellogenin (control)	$2 \cdot 10^{-5}$	-	-		pearson
S2e	Apolipoprotein 3 (control)	0.12	-	-		pearson
S2e	Vitellogenin (D4)	$1.3 \cdot 10^{-4}$	-	-		pearson
S2e	Apolipoprotein 3 (D4)	0.024	-	-		pearson
S4d	Control	$<2.2 \cdot 10^{-16}$	-	-		pearson
S4d	D4	$<2.2 \cdot 10^{-16}$	-	-		pearson
S4e		$<2.2 \cdot 10^{-16}$	-	-		pearson

S5a	DNMT1 (Control)	0.26	-	-		pearson
S5a	DNMT1 (D4)	0.25	-	-		pearson
S5a	DNMT2 (Control)	0.4	-	-		pearson
S5a	DNMT2 (D4)	0.16	-	-		pearson
S5a	TDG long (Control)	0.25	-	-		pearson
S5a	TDG long (D4)	0.78	-	-		pearson
S5a	TDG short (Control)	0.065	-	-		pearson
S5a	TDG Short (D4)	0.94	-	-		pearson
S5a	TET (Control)	0.057	-	-		pearson
S5a	TET (D4)	0.46	-	-		pearson
S5a	UHFR1 (Control)	0.15	-	-		pearson
S5a	UHFR1 (D4)	0.016	-	-		pearson
S5a	UNG (Control)	0.24	-	-		pearson
S5a	UNG (D4)	0.58	-	-		pearson
S5b	DNMT1 (Control)	0.39	-	-		pearson
S5b	DNMT1 (D4)	0.66	-	-		pearson
S5b	DNMT2 (Control)	0.78	-	-		pearson
S5b	DNMT2 (D4)	0.049	-	-		pearson
S5b	TDG long (Control)	0.36	-	-		pearson
S5b	TDG long (D4)	0.3	-	-		pearson
S5b	TDG short (Control)	0.93	-	-		pearson
S5b	TDG short (D4)	0.066	-	-		pearson
S5b	TET (Control)	0.0072	-	-		pearson
S5b	TET (D4)	0.066	-	-		pearson
S5b	UHFR1 (Control)	0.27	-	-		pearson
S5b	UHFR1 (D4)	0.84	-	-		pearson
S5b	UNG (Control)	0.24	-	-		pearson
S5b	UNG (Control)	0.84	-	-		pearson

In Fig. 2b we now list all raw p-values tested (along with the corrected p-values in the table) such that the reader can gain an impression about both false positive and false negative rates.

I also would like to see a discussion of why the FDR correction was changed from 0.05 to 0.1. This is regrettably often done, and sometimes there are good reasons, but without explaining why this choice was made it is hard to judge this with confidence. So more transparency would help a lot, e.g. what would the statistical analysis have shown in FDR was controlled at 0.05 throughout?

In our manuscript the FDR is always controlled at a level $\alpha=0.05$. When selecting significantly variable genes we chose a threshold of 0.1 which slightly affects Fig. 4g and the significance level of the overrepresentation of differentially expressed queen genes reported in the main text. In Fig. 4g, the threshold was set to identify a list of highly variable genes. The results depend only weakly on the chosen value of the threshold (from left to right its value is 0.1, 0.05, 0.01) as:

The associated p-values for the hypothesis that the average fraction of variable genes is identical between the Control and D4 phases are 2.1×10^{-8} ($\alpha=0.1$), 2.5×10^{-8} ($\alpha=0.05$) and 1.2×10^{-9} ($\alpha=0.01$), respectively.

Further, the overrepresentation of queen genes in the set of DEGs does not change significantly when adjusting the FDR value, namely $p=1.6 \times 10^{-36}$ for $\alpha=0.1$, $p=1.5 \times 10^{-70}$ for $\alpha=0.05$ and $p=4.1 \times 10^{-31}$ for $\alpha=0.01$.

Taken together, as our results do not depend on the choice of significance level, we now report the results obtained for a threshold of 0.05 in the revised version of the manuscript. We also now explicitly state that FDR correction was controlled for $\alpha=0.05$ in the methods section.

In general I found the statistical analysis not reproducible to the extent that I would like to be able to.

We apologise for not having included sufficient details for full reproducibility of our statistical analysis. We extended the STAR methods section, figure legends and included a supplementary table containing all information needed to reproduce the statistical analysis.

- I could not find sufficient detail on the statistical analysis of the video analysis. In particular I would like to see an assessment of classification/mis-classification rates. The data in Fig 2b is hard to see and interpret. Are the p-values raw or corrected?

Two types of analyzes were performed on the videos: quantification of overall nest activity using computer vision and a classification of wasp behaviours. Only the former was automated by a machine learning program. The classification of wasp behaviours into 6 categories was performed manually, reducing the possibility of mis-classification compared to computational approaches. We have now specified this information in the methods section. In order to estimate the uncertainty associated with the manual classification of wasp behaviours we repeated the manual classification. The original analysis was performed in 2018. This new analysis was consistent with the previous analysis. We observed roughly 15% (157 versus 128) more interactions in total, which could be attributed to the use of updated hard- and software which allowed for better slowdown of the video playback. Significance levels were largely unaltered and we now show the newer analysis in Fig. 2.

More generally, we listed all the details of the statistical tests, the p-values and the corrected p-values (including those in Fig2b) in a new supplementary table. Finally, we have improved the visualisation of the vertical separation of data points in Fig. 2b. In Fig. 2b we now also show all unadjusted p-values.

- The theoretical modelling is very well explained in the supplementary information. This is very nicely written up. The discussion in the paper is, however, too terse and not complete. For example, the purely additive Ansatz for the change in probability flux of molecular and colony level processes is much better explained in the SI where the absence on interactions is clear as an assumption. I would like to see the same transparency in the main paper, too.

In the revised version of the manuscript, we added details on how in the model interactions lead to a change in queen gene expression rates and how the approximations conducted in mathematical detail in the Supplemental Theory lead to the additive form presented in the main text (p. 9).

- The non-Markovian dynamics are also nicely derived in the supplementary information. The discussion in the main paper is cryptic and not helpful. Only a fraction of readers would find "Hartree approximation" helpful, I am sure, whereas the more common terminology (also used in the SI throughout) "meanfield" is clearer and to most people more familiar.

We now elaborate more on the derivation of the non-Markovian dynamics in the main text on p. 9.

- Supplementary Figure 2g and h strikes me as encouraging regarding the modelling efforts. Why are these results relegated to the supplementary material?

We followed the reviewer's advice and moved these figures to the main figures, Fig. 2.

Reviewer #2:

This manuscript presents a model, supported by experimental evidence, for how paper wasp colonies maintain a stable social structure while also being able to rapidly reconfigure in response to external perturbation. Colonies have a stable division between a single reproductive queen and dozens of non-reproductive workers. This study posits that all workers have a constitutive tendency to become queens, but this development is inhibited by queen-initiated dominance interactions. If the queen is removed, all workers begin to develop a queen-like phenotype, resulting in an increase in dominance interactions and a subsequent re-repression of queen development in most workers. Repression is posited to result from a tendency of dominance interactions to suppress queen phenotype in losers of the interactions.

Most of the components of this model seem reasonable. I think many of the basic ideas about self-activation and mutual inhibition have been put forward in earlier work. However, a strength of the study is the combined behavioral, physiological, and molecular evidence for rapid emergence of a queen-like phenotype in large numbers of workers after queen removal.

Another is the development of a credible dynamical model showing how the basic assumptions of the model can lead to the combination of stability and plasticity observed in the experiments. This model shows that perturbations in noisy caste determination mechanisms within single wasps will be damped out in a queenright colony; only if queen removal sharply lowers the overall amount of queen-determining factors will the colony enter a regime where changes in caste become possible. This combination of dynamic modeling and rich evidence at multiple levels of organization is a novel and promising advance in the study of these societies.

We thank the reviewer for their positive evaluation of our work.

The section on the role of DNA methylation muddles the picture somewhat. After describing how the dynamical system of the colony creates stable worker and queen phenotypes, the paper then gives results contradicting this stability; namely, chance activation of queen genes is said to lead to an expected queen persistence time of only a few days. The authors then go on to present interesting evidence for how DNA methylation might make the social structure stable in the face of such perturbations. However, the paper should be modified to explain how this instability is consistent with the apparently contradictory model results previously described and summarized in Figure 3.

The model discussed in Figs. 2 and 3 are mean-field analyses. Briefly, the mean-field analysis predicts that the colony is stable against intrinsic noise in the sense that the steady state is metastable with respect to perturbations which are uncorrelated across insects. The precise time scale of escape from this metastable state depends on the specific rates describing the dynamics on the molecular scale and cannot be rigorously deduced from this mean-field analysis.

In the section about the role of DNA methylation we calculate the quantitative time scale of the nest stability taking into account strong noise and from this we deduce statistical properties of the biological molecular processes encoded in the variable r .

We do this by constructing an argument by contradiction. We start with the hypothetical assumption that the variable r represents gene expression dynamics (or molecular dynamics on similar time scales) alone. Using this assumption we deduce a contradiction by predicting from typical rates of gene expression and protein degradation a typical life-time of the colony that is much shorter than what is observed in experiments. As this estimate is largely independent of our model assumptions we conclude that our hypothetical assumption, namely that the variable on the molecular scale represents pure gene expression dynamics, is false. The role of the derivation of this paradox is to motivate additional processes influencing the variance of gene expression in the following paragraphs.

Due to the use of this argument by contradiction this section could easily be misunderstood to be a contradiction to the previous sections of the manuscript. To clarify this point we now make the logical structure of the section about the role of DNA methylation and its relation to the previous paragraph clear in the revised version of the manuscript.

It would also help if the paper was clearer about which aspects of the model have experimental evidence and which do not. For example, I did not see evidence for the key idea that losing a dominance interaction reverses the queen-like molecular, physiological, and behavioral phenotype.

The effect of an absence of dominance interactions is not an explicit assumption in the model. Rather, the absence of interactions tilts the balance between the remaining processes affecting queen-like molecular signatures. That said, there is evidence for our explicit model assumption that dominance interactions lead to a repression of queen-like phenotypes in the subdominant individual. In particular, in experimental work done with *Polistes gallicus* it was found that that when a hierarchy is established after dominance behaviours, the endocrine activities of subordinate individuals are inhibited by the reproductive queen (Röseler, P.F., et al. *Behav Ecol Sociobiol* 15, 133–142 (1984)). Although the field

conditions under which our study was conducted did not allow us to directly reveal this endocrine repression, the re-establishment at D14 of a weak correlation with the expression of the queen genes (Fig.1f), the absence of individuals with developed ovaries (Sup. Fig. 2g) and the decrease in aggressive behavior (Fig 2b) shows a reversion at the molecular, physiological and behavioral levels of the phenomena initiated in the workers just after the removal of the queen. We experimentally observed a constitutive expression of queen genes after queen removal (Fig.1d,e,f) which is further corroborated by previous studies on *Polistes* and other social insects (Taylor, B.A., et al., *Nat Commun* 12, 775 (2021); Oi, CA et al., *Front Ecol Evol* 7, (2019)). To address the reviewers comment we revised the main text to be clearer about which aspects of the model are based on experimental observations (pp. 9,10).

The authors might consider re-writing the abstract to be less generic. I understand the need to communicate the generality of their results, but it is very hard to tell from the abstract what is the actual phenomenon under study (regulating queen number in wasp colonies) or the basic idea of how it works (a dynamic balance between constitutive queen development in all colony members and inhibition of queen development by established queens).

We rewrote the abstract following the reviewer's advice. Please note that queen establishment is an emergent phenomenon in the model. In principle all insects can interact with each other, but with rates that are proportional to their molecular state. As Reviewer 1 pointed out, this aspect was only clearly defined in the supplement, but not in the main text, which we corrected in the revised version of the manuscript.

The authors might consider exploring whether their model can account for a greater variety of social structures than the three they show in Figure 2e. For example, can it account for a linear hierarchy or the oligarchies seen in certain ants? They might have a look at the paper by Sasaki et al. (*American Naturalist*, 2016) that presents a very different kind of model of related questions.

This is an interesting question. Indeed, our model can describe multiple population structures by suitably tuning the model parameters. The steady-state distribution of queen gene expression across the population is modified in response to changes in the model parameters, as reflected by changes in the variance of the distribution, the number of modes and their position. For instance, our model recapitulates the shared hierarchies introduced in the Sasaki et al. paper without the need to account for winner-winner interactions. However, in Fig. 2e we chose to summarize the steady-state distribution of queen gene expression across the population using the average queen fraction observed in the simulations for the sake of clarity, leading to the definition of the three regimes mentioned by the reviewer. Furthermore, although in this paper we focus on one specific form of the interaction kernel motivated by biological observations, different choices of the interaction kernel modify the number of fixed points of the mean field equation, offering the possibility to tune the number of stable phenotypes by choosing a suitable interaction term. We also cited in the discussion the paper mentioned by the reviewer.

Some of the figure legends could use more explication. For example, they could better describe the meaning of the dashed blue and gray lines in Figure 3a and give a quick definition of fight index and dominance index in Figure 2b. In Figure 2e, it's hard to understand why there is a Non-reproductive phase. Does this mean there are no queens? The color coding does not suggest any fewer queens there than in the Social zone. In Figure 1e, why is one of the arrows in the radar plot black?

We revised the figure legends to incorporate the information the reviewer requested, Specifically,

- Fig. 3a: The gray line corresponds to the steady state average queen gene expression across the population. The dashed blue line corresponds to the initial average queen gene expression of a hypothetical queenless population. The solid blue line represents the temporal evolution of the queen gene expression of a particular individual from this

population. In the left and right panels pink lines denote perturbation and blue lines the ensuing relaxation dynamics. We added this to the figure legend.

- Fig. 2b: We added the definition of fight and dominance indices to the figure legend.
- Fig. 2e: There are indeed no queens in this phase. In this hypothetical parameter regime no insect is able to achieve sufficient dominance to become queen. As the colour code gives the fraction of queens in a society of typical size it is difficult to distinguish between 0 and 1 queens, which is why we now explicitly state the number of queens in the respective regimes.
- Fig. 1e: Color in the bottom panel indicates the PCA dimension, such that blue arrows signify the strongest contributions to PC1 and the black arrow represents the gene with the strongest contribution to PC2. Shown are the top 20 genes, only Queen genes are named in the figure. We have added this information to the figure legend in the revised version of the manuscript.